# Reduced Absolute Count of Monocytes in Patients Carrying Hematological Neoplasms and SARS-CoV2 Infection

**DOI:** 10.3390/cancers14051173

**Published:** 2022-02-24

**Authors:** Alessandra Romano, Claudio Cerchione, Concetta Conticello, Sabina Filetti, Anna Bulla, Annalisa Chiarenza, Vittorio Del Fabro, Salvatore Leotta, Uros Markovic, Giovanna Motta, Marina Parisi, Fabio Stagno, Giuseppe Alberto Palumbo, Francesco Di Raimondo

**Affiliations:** 1Division of Hematology, AOU “Policlinico G. Rodolico-San Marco”, 95100 Catania, Italy; sandrina.romano@gmail.com (A.R.); ettaconticello@gmail.com (C.C.); filettisabina2@gmail.com (S.F.); anna.bulla24@gmail.com (A.B.); annalisa.chiarenza@gmail.com (A.C.); vdelfabro@yahoo.it (V.D.F.); leotta3@yahoo.it (S.L.); urosmarkovic09041989@gmail.com (U.M.); gmotta72@gmail.com (G.M.); marinaparisi@hotmail.it (M.P.); fsematol@tiscali.it (F.S.); palumbo.giuseppealberto@gmail.com (G.A.P.); diraimon@unict.it (F.D.R.); 2Postgraduate School of Hematology, University of Catania, 95124 Catania, Italy; 3Hematology Unit, Istituto Scientifico Romagnolo per lo Studio e la Cura dei Tumori [M1] (IRST) IRCCS, 47014 Meldola (FC), Italy; 4Oncohematology and BMT Unit, Mediterranean Institute of Oncology, 95125 Viagrande, Italy; 5Department of Biomedical, Dental, Morphological and Functional Imaging Sciences, University of Messina, 98122 Messina, Italy; 6Ingrassia Department, University of Catania, 95100 Catania, Italy

**Keywords:** COVID-19, monocytes, hematological malignancies, COVID-19 serological testing, multiple myeloma, breakthrough COVID-19

## Abstract

**Simple Summary:**

In hematological neoplasms associated with COVID-19, immunological dysfunction, including reduced count of non-classical monocytes, has been suggested as a primary driver of morbidity and mortality. In this work, we investigated the contribution of absolute monocyte count to clinical outcome of COVID-19 in 120 patients affected by hematological neoplasms that tested positive to SARS-CoV-2. We found that there was no statistical difference in 30-day mortality, rate of hospitalization for intensive cure and viral clearance at 14 days between fully vaccinated and unvaccinated patients. Increased 30-day mortality was associated with presence of active/progressing disease and absolute monocyte count lower than 400 cells/uL. Reduced absolute counts of monocytes should be used as an alert of increased risk of severe/critical forms of COVID-19 in patients with hematological malignancies, even when the full vaccination cycle has been completed.

**Abstract:**

Background: Clinical course of COVID-19 depends on several patient-specific risk factors, including immune function, that is largely compromised in cancer patients. Methods: We prospectively evaluated 120 adult consecutive patients (including 34 cases of COVID-19 breakthrough after two full doses of BNT162b2 vaccine) with underlying hematological malignancies and a SARS-CoV-2 infection, in terms of patient’s clinical outcome. Results: Among fully vaccinated patients the achievement of viral clearance by day 14 was more frequent than in unvaccinated patients. Increased 30-day mortality was associated with presence of active/progressing disease and absolute monocyte count lower than 400 cells/uL. Results of multivariable analysis in unvaccinated patients showed that the pre-infection absolute count of monocytes less or equal to 400 cells/mmc, active or progressive disease of the underlying hematological malignancy, the COVID-19 severity identified by hospitalization requirement and lack of viral clearance at 14 days were independent predictors of 1-year overall survival. Conclusions: Taken together, our results indicate that absolute monocyte count determined one month before any documented SARS-CoV-2 infection could identify patients affected by hematological neoplasms with increased risk of inferior overall survival.

## 1. Introduction

COVID-19 (coronavirus disease-2019) is a complex disease with variable clinical presentations and outcomes, due to the infection of a novel β-Coronavirus SARS-CoV-2 [1]. In most cases, COVID-19 symptoms are moderate or totally absent, with about one week of incubation period. Around 15% of patients can progress to severe pneumonia and about 5% eventually progress to acute respiratory distress pneumonia, renal failure, septic shock, multiple organ failure and death [2,3,4]. 

Due to the heterogeneous clinical course of COVID-19, several biomarkers have been evaluated that could allow us to predict an initial severe presentation or critical evolution of the disease. Several observational studies have suggested that some comorbidities, assessed by high Charlson comorbidity index (CCI ≥ 3) scores are disproportionately associated with inferior clinical outcomes [5].

Clinical course and disease severity in COVID-19 is strongly associated with weaker immune response, bulk release of proinflammatory cytokines and the recruitment of neutrophils, monocytes and macrophages, which can generate an aggressive response, in some cases inappropriate, detrimental and harmful for the host [6], especially in cancer patients [7,8]. Recent reports disclosed the contribution of monocytes to the hyper-inflammatory phenotype, thus worsening disease severity [9,10,11]. In vitro, SARS-CoV-2 can induce a functional specialization of dendritic cells subsets, with high levels of interferon-α, IL-6, IL-10 and IL-8 that orchestrate and propagate first the innate and then the adaptive immune response [8,12,13]. High levels of IL-6 can further trigger cytokine storm in the absence of appropriate type I and III interferon response [14,15]. 

Patients carrying hematological neoplasms have increased infection susceptibility, due to immunodeficiency, T-cell anergy, increased myeloid-derived suppressor cells and impairment of antigen presentation machinery [6,13,16], as a consequence of the malignancy itself. In multiple myeloma (MM), either anergic, dysfunctional effector lymphocytes or both, tumor-educated myeloid-derived suppressor cells and soluble mediators promote coordinately cancer immune-evasion [17,18,19,20,21,22]. In chronic lymphatic leukemia (CLL), circulating monocytes have a skewed aberrant phenotype leading to altered composition and phagocytosis, contributing T-cell exhaustion and anergy [23,24,25,26]. In Hodgkin’s lymphoma (HL) monocyte-like myeloid derived suppressor cells are increased and correlate to chemosensitivity [19,27,28,29,30,31]. In acute myeloid leukemia (AML), macrophages and myeloid-derived cells are educated by leukemia itself to develop a supportive phenotype, and thus, contribute to drug resistance [32,33,34,35].

However, little is known about the contribution of perturbed counts and functions of monocytes to COVID-19 clinical course in patients carrying hematological malignancies, and if the perturbed immune parameters observed in critically ill COVID-19 patients retain the same significance in the setting of patients affected by hematological neoplasms. To this end, we designed a single-center prospective study, to determine the contribution of monocyte accounting in the clinical outcome of 120 consecutive patients affected by hematological neoplasms and tested positive to SARS-CoV-2 in our center from 15 April 2020 through 30 November 2021. 

## 2. Materials and Methods

### 2.1. Patients’ Selection

Our study included 120 adult consecutive patients (aged ≥18 years) with a WHO-defined hematologic malignancy that tested positive to SARS-CoV-2 in the emergency departments, hospital wards (patients infected while hospitalized) or outpatient clinics of the Division of Hematology, AOU Policlinico in Catania, Sicily, Italy from 15 April 2020 through 30 November 2021. Patients were categorized as fully vaccinated at the time of COVID-19 when two doses of vaccine BNT162b2 had been administered and diagnosis of COVID-19 was recorded >4 weeks from the last dose, thus identifying the breakthrough infections after COVID-19 vaccination. Unvaccinated patients were defined as having no known prior exposure to COVID-19 vaccination before COVID-19 diagnosis.

The COVID-19 diagnosis was confirmed by nasopharyngeal swab collection in accordance with local prevention guidelines. The study was approved by the Institutional Review Board (IRB) (Comitato etico Catania 1, https://www.policlinicovittorioemanuele.it/comitato-etico-catania-1 (accessed on 15 April 2020), #CO.TIP. 34/2020/PO 0,016,693 released on 15 April 2020) and performed in accordance with the principles of the Declaration of Helsinki and the International Conference on Harmonization Good Clinical Practice guidelines.

### 2.2. Procedures

Electronic health records of patients followed in our hospital were evaluable to capture the following information: type of hematological malignancies, details and timing of the tumor treatment, laboratory parameters at the time of infection, outcome of the SARS-CoV-2 infection and outcome of the hematological malignancies at the time of last follow-up. Active antineoplastic treatment was defined as having received anticancer therapy within 30 days prior to COVID-19 diagnosis. Infection laboratory values were collected no more than 7 days preceding the first documentation of SARS-CoV-2 infection. COVID-19-related death was categorized in accordance with the WHO definition, while comorbidity was classified according to the modified Charlson comorbidity index (CCI).

In unvaccinated patients, seroconversion was performed on serum samples to detect human antibodies of the immunoglobulin classes IgG and IgA against the SARS-CoV-2 by anti-SARS-CoV-2 ELISA IgG and IgA assays (Euroimmun), until one month from documented viral clearance, according to the manufacturer’s instructions. 

In vaccinated patients, the titer of antibodies developed against the receptor-binding domain of the SARS-CoV-2 spike protein was measured 30 days after the second dose of BNT162b2 vaccine using the SARS-CoV-2 IgG II Quant assay (Abbott, CE marked), by chemiluminescence (CMIA) method, performed on the Abbott Alinity i platform according to the manufacturer’s instructions. 

### 2.3. Statistical Analysis

Continuous variables were expressed as median and range (minimum–maximum), since a preliminary analysis showed that data distribution was not normal. Normality was verified using the Shapiro–Wilk test and graphically using Q–Q plot. Counts and percentages of qualitative variables were generated for descriptive statistical analysis. For further comparisons, we used the Mann–Whitney U test for continuous data and the Fisher’s exact test for categorical data. 

We applied propensity score matching (PSM) as a consequence of the limited sample size and its heterogeneity to adjust for differences in baseline clinical variables between fully vaccinated and unvaccinated patients [8]. The covariates balanced between groups were: age (used as a dichotomic variable, less than 70 years, equal to or more than 70 years old), biologic sex (female; male), Eastern Cooperative Oncology Group performance status (ECOG PS 0–1; ≥2), lymphopenia (absolute lymphocyte count (ALC), ≤1000 vs. >1000 per μL), Charlson comorbidity index (0–1; vs. ≥2), cancer status (active and progressing vs. not active and progressing), hematological neoplasm type (lymphoid; myeloid or plasma cell neoplasm). In PSM, we selected the caliper 0.25 of the standard deviation for drawing the control units (unvaccinated) to match the treated units (fully vaccinated) with the nearest-neighbor method and a 3:1 ratio (unvaccinated:fully vaccinated). Due to limited number of events, we considered variable selection in regression analysis by elastic-net regularization with a mixing parameter (LASSO). Following LASSO variable selection, the additional inclusion of AMC < 400 cells/uL was considered given a significant association with the primary endpoint and prior evidence suggesting the association of this variable to immune dysregulation in COVID-19 patients carrying hematological neoplasms [6]. 

The primary endpoint was 30-day all-cause mortality (infection, progressive disease, other) among fully vaccinated patients affected by hematologic malignancy who tested positive to SARS-CoV-2 compared to the cohort of unvaccinated hematological patients after PSM adjustment for baseline clinical variables. Secondary endpoints included rates of hospitalization in intensive care units and viral clearance at 14 days, in fully vaccinated, compared with unvaccinated patients with hematological neoplasms after PSM adjustment for baseline clinical variables.

All calculations were performed using MedCalc Statistical Software version 13.0.6 (MedCalc Software bvba, Ostend, Belgium; http://www.medcalc.org (accessed on 6 January 2022); and XLSTAT version 2021.5-Life Sciences, released in December 2021.

## 3. Results

### 3.1. Characteristics of the Cohort

We diagnosed COVID-19 infection in 120 patients affected by hematological neoplasms, of whom 86 (72%) were unvaccinated and 34 (28%) were fully vaccinated, including 18/34 (15%) patients who had received a boost dose within 2 weeks from COVID-19 diagnosis. 

Baseline characteristics of the cohort are summarized in Table 1. The median age was 65 years (range 23–94 years) in the unvaccinated group, without any significant difference to fully vaccinated patients. In the fully vaccinated group, most patients were receiving active treatment for their hematological malignancy, differently from the unvaccinated group (*p* < 0.0001), mostly consisting of immunotherapy and targeted therapy, like lenalidomide maintenance in patients affected by multiple myeloma. The median value of Charlson comorbidity index was 3 (0–14), in the unvaccinated group, with the highest score among patients affected by lymphoid neoplasms, due to cardiovascular disease, pulmonary disease and diabetes (data not shown). A Charlson comorbidity index equal to or higher than 2 was more frequent in the unvaccinated group than in vaccinated patients (*p* = 0.003, Table 1).

There were no significant differences in pre-infection laboratory parameters evaluated among unvaccinated and fully vaccinated patients, including absolute counts of neutrophils, monocytes and lymphocytes, C-reactive protein (C-RP) and lactate de-hydrogenase, which have been associated with COVID-19 severity in both general population and cancer subjects [36,37].

### 3.2. Outcome of COVID-19 in Patients Affected by Hematological Neoplasms 

Among fully vaccinated patients the achievement of viral clearance by day 14 was more frequent than in unvaccinated patients (*p* = 0.0003, Table 2). Only one fully vaccinated vs. 22 (26%) unvaccinated patients was admitted for inpatient intensive care (*p* = 0.004, Table 2), due to concomitant neutropenic fever and bacterial pneumonia. 

Unvaccinated patients who achieved viral clearance by 14 days from the documented SARS-CoV-2 infection had a lower Charlson comorbidity score, due to significant lower frequency of heart disease (32%, *p* = 0.04, data not shown). There was no significant difference among patients affected by myeloid, lymphoid or plasma cell neoplasm, active or not (data not shown). However, the number of patients who died from COVID-19 could not reach any significant difference among fully and unvaccinated patients (Table 2). 

Unvaccinated patients who required hospitalization for their COVID-19 in an intensive care unit were older than those who did not require hospitalization (*p* = 0.002), with higher median Charlson comorbidity index (5 vs. 3), due to significantly higher frequency of heart disease (*p* = 0.01). There was no significant difference among patients affected by myeloid, lymphoid or plasma cell neoplasm, active or not. However, there were more patients in treatment with chemotherapy among hospitalized patients (*p* = 0.04), while non-hospitalized patients received targeted therapy more frequently (*p* = 0.004), reflecting the higher probability of hospitalization for those with chemotherapy-related immune deficiency. Patients requiring hospitalization for COVID-19 had lower hemoglobin (*p* = 0.03) and absolute monocyte counts (*p* = 0.02, Table 3).

The median value of anti-SARS-CoV-2 antibodies (IgG) titer was 4.7 (range 1.1–38.7 BAU), as measured at one month from the documented viral clearance in 66/86 unvaccinated patients, showing that 42/66 (64%) patients were seroconverted.

Among the fully vaccinated patients, the anti-SARS-CoV-2 antibodies (IgG) titer was 5.2 (range 0–13.1 BAU) in 24/34 tested patients, showing that none of them achieved a protective titer (>40 BAU) at one month after two doses of BNT162b2 vaccine. 

Following PSM there was no statistical difference in 30-day mortality, rate of hospitalization for intensive cure and viral clearance at 14 days between fully vaccinated and unvaccinated patients, as shown in Table 4 where adjusted odds ratios (AOR) and 95% confidence intervals (CI) have been reported. Increased 30-day mortality was associated with the presence of active/progressing disease and absolute monocyte count lower than 400 cells/uL (presence of non-active disease vs. active/progressing disease: AOR −0.64, 95% CI: −1.09–0.2; absolute monocyte count (AMC) less than 400 cells/uL vs. AMC ≥ 400 cells/uL, AOR 0.68, 95% CI: 0.14–1.21). 

### 3.3. Pre-Infection Biomarkers Associated to Inferior Overall Survival in Unvaccinated Patients with Hematological Malignancy and SARS-CoV-2 Infection

We then assessed potential biomarkers for 1-year overall survival (OS) of patients carrying hematological neoplasms that tested positive for SARS-CoV-2 infection. To this end, we considered only the cohort of unvaccinated patients due to their longer median follow-up (13.6 months).

In unvaccinated patients of our series, the median OS was 10.4 months (95% CI. 9.6–11.3), which was affected in univariate analysis, summarized in Table 5, by presence of: active disease (*p* = 0.003), pre-infection absolute count of monocytes less or equal to 400 cells/mmc (*p* = 0.04), hospitalization due to COVID-19 (*p* < 0.0001), lack of viral clearance at 14 days (*p* = 0.005) and lack of seroconversion (*p* = 0.04). The difference in survival was consistent throughout all subgroups tested, independent of whether the patients were aged below or above 70 years, of female or male sex, or suffered from myeloid, lymphoid or plasma cell neoplasm. 

Results of multivariable analysis of OS showed that the pre-infection absolute count of monocytes less or equal to 400 cells/mmc (*p* = 0.008), active or progressive disease of the underlying hematological malignancy (*p* = 0.009), COVID-19 severity identified by hospitalization requirement (*p* = 0.004) and lack of viral clearance at 14 days (*p* = 0.03) were independent predictors of 1-year OS in unvaccinated patients affected by hematological neoplasms tested positive for SARS-CoV-2 infection (Table 6).

## 4. Discussion

Several studies showed higher risk of death in cancer patients with COVID-19, along with higher rates of admission in the intensive care unit and development of severe complications from COVID-19 [4,37,38,39,40,41,42,43,44]. Worst outcomes, with death rates of 18.18% and 33.33%, were registered respectively among lung and blood cancer patients [45,46]. 

In our series of 120 patients affected by hematological neoplasms, the outcome was poor, with higher frequency of hospitalization and death compared to data obtained from general settings, confirming the first report published in August 2020, based on a multi-center series of 536 hematological patients with COVID-19 [46]. However, breakthrough COVID-19, occurring in those patients who developed COVID-19 despite full vaccination cycle in our series was mild, being associated with a lower rate of hospitalization and higher frequency of early viral clearance at 14 days. Preliminary reports show that patients with hematological neoplasms have increased risk of developing breakthrough COVID-19 following full vaccination and remain susceptible to severe outcomes [47].

In patients with severe COVID-19, significant and global alterations in both T-, B- and myeloid cell compartments have been described, and this underlines the immune system’s effort to make up for lymphopenia and loss of naïve T-cells by recruiting switched B-memory cells, besides the cytokine storm and the functional and phenotypic alterations of the innate response compartment [6,13,48].

The inappropriate response to the virus in patients with moderate and severe disease is the result of a complex interaction between virus, host and environment which affect entry, replication, egress and innate immune control [9,49]. In our series we could not find any correlation between clinical severity or overall survival after COVID-19 associated to changes in absolute counts of neutrophils. Differently from data obtained from the general population or in cancer patients, the pre-infection absolute count of neutrophils and lymphocytes, C-RP and LDH were not associated to OS in unvaccinated patients included in our series [36,46]. However, C-RP and LDH were higher in those patients who did not achieve viral clearance by 14 days, reflecting the impairment of the inflammatory status in COVID-19 [50]. 

Surprisingly, patients with inferior outcome after COVID-19 had lower absolute monocyte count, differently from previous reports in the general population, where an impairment in monocyte counts and function has been largely described [9,10,51,52]. Indeed, large studies on the general population showed that patients in severe-stages of COVID-19 had increased amounts of circulating CD14^+^ CD16^+^ monocytes which exert inflammatory activity through increased release of IL-6 and interaction with adaptive B and T-cells [53,54]. However, decreased frequencies of non-classical monocytes, as consequence of dysregulated emergency myelopoiesis, has been proposed to discriminate patients who develop a severe form of COVID-19 [55,56]. Compared to COVID-19 patients without hematological cancer, patients carrying hematological neoplasms have decreased percentages of classical monocytes, immunoregulatory natural killer cells, double-positive T cells and B cells, that could compromise an initial response to the infection [6].

Despite the limited number of patients involved in our work, our data confirm the impairment in the viral clearance in patients affected by hematological neoplasms, and the importance of early viral clearance on clinical outcome and overall survival. Patients who did not develop seroconversion after COVID-19 vaccination had higher probability of achieving viral clearance by 14 days, an independent predictor of overall survival at one year for unvaccinated patients [2,50,57].

Thus, if on one hand patients carrying hematological neoplasms had increased risk of not developing a serological response against COVID-19 infection [58,59] or vaccine [60,61,62,63,64], due to treatment-mediated immune dysfunction, on the other hand they can improve viral clearance if fully vaccinated [65,66]. This hypothesis needs to be tested in larger, multi-center cohorts, to explore the clinical course of COVID-19 breakthroughs [47] and identify an effective prevention strategy in contrasting severe/critical forms of COVID-19 in patients with hematological malignancies.

## 5. Conclusions

Taken together, our results indicate that absolute monocyte count determined one month before any documented SARS-CoV-2 infection could identify among patients affected by hematological neoplasms those with increased risk of inferior overall survival at one year, due to increased risk of hospitalization, lack of seroconversion and viral clearance at 14 days.

## Figures and Tables

**Table 1 cancers-14-01173-t001:** Pre-infection clinical and laboratory parameters in hematological patients affected by COVID-19 by baseline vaccination status.

Pre-Infection Clinical and Laboratory Parameters	Fully Vaccinated (N = 34)	Unvaccinated (N = 86)	*p*-Value ^a,b^
Median age, years (range)	63 (30–94)	65 (23–94)	0.87
Female gender, N (%)	12 (35)	43 (50)	0.13
Active and progressive disease, N (%)	12 (35)	27 (31)	0.93
Systemic treatment within 3 months, N (%)	32 (94)	45 (52)	** *<0.001* **
ALC, ×10^3^ cells/uL (range))	1.26 (0.4–188.0)	1.5 (0.03–79.5)	0.32
ANC, ×10^3^ cells/uL (range)	3.7 (0.2–15.2)	3.5 (0.0–22.1)	0.98
AMC, ×10^3^ cells/uL (range)	0.5 (0.1–2.1)	0.4 (0.01–1.7	0.98
LDH, IU/L (range)	208 (92–980)	201 (112–5000)	0.97
C-RP, IU/L (range)	5 (1–157)	2 (0–126)	0.12
Hematological malignancy type			
Lymphoid neoplasm, N (%)	8 (24)	29 (34)	0.38
Myeloid neoplasm, N (%)	6 (18)	21 (24)	0.48
Plasma cell neoplasm, N (%)	20 (59)	36 (42)	0.09
Charlson comorbidity index 2 or more, N (%)	16 (47)	66 (75)	** *0.003* **
ECOG performance status 2 or more, N (%)	10 (30)	27 (31)	0.92

^a^ based on Fisher-exact test, ^b^ based on Mann–Whitney test. Abbreviations: ALC, absolute lymphocyte count; AMC, absolute monocyte count; ANC, absolute neutrophil count; uL, micro-liter; IU/L, international units per liter; LDH, lactate de-hydrogenase; C-RP, C-reactive protein; ECOG, Eastern Cooperative Oncology Group.

**Table 2 cancers-14-01173-t002:** Clinical outcomes of COVID-19 in hematological patients by baseline vaccination status.

COVID-19 Infection Severity and Clinical Outcome	Fully Vaccinated (N = 34)	Unvaccinated (N = 86)	*p*-Value ^a^
Number of patients (%) with viral clearance on day 14	28 (82)	39 (45)	** *0.0003* **
Number of patients (%) who required hospitalization for intensive care	1 (3)	22 (26)	** *0.004* **
Number of deaths (%) by day 30 from COVID-19 diagnosis	1 (3)	10 (12)	0.13

^a^ based on Fisher-exact test.

**Table 3 cancers-14-01173-t003:** Pre-infection clinical and laboratory parameters in hematological patients hospitalized or not hospitalized for concomitant COVID-19.

Clinical and Laboratory Parameters Prior to COVID-19 Infection	Hospitalized	Not Hospitalized	*p*-Value ^a,b^
Patients, N (%)	22 (26)	64 (74)	
Age, years median (range)	71 (31–87)	63.5 (24–94)	** *0.02 ^b^* **
Sex, n (%)			
Female	8 (36)	37 (55)	0.14 ^a^
Male	14 (66)	27 (45)	
Charlson comorbidity index			
Median (range)	5 (0–13)	3 (0–14)	NA
Coexisting condition, N (%)			
Heart disease	10 (46)	12 (19)	** *0.01 ^a^* **
Pulmonary disease	7 (23)	9 (14)	0.32 ^a^
Vascular disease	7 (23)	10 (16)	0.46 ^a^
Connective tissue disease	1 (5)	0 (0)	0.26 ^a^
Liver disease	1 (5)	0 (0)	0.2 ^a^
Diabetes	6 (27)	8 (13)	0.13 ^a^
Non-hematological cancer	1 (5)	5 (9)	0.98 ^a^
Type of hematological malignancies, N (%)			
Myeloid neoplasms	4 (18)	16 (25)	0.56 ^a^
Lymphoid neoplasms	9 (41)	20 (32)	0.60 ^a^
Plasma Cell Neoplasms	9 (41)	28 (44)	0.99 ^a^
Active therapy, N (%)	10 (46)	18 (30)	0.20 ^a^
Last therapy type, N (%)			
Chemotherapy	9 (41)	11 (18)	** *0.04 ^a^* **
Immunotherapy	0	1 (2)	0.98 ^a^
Targeted therapy	1 (5)	9 (14)	** *0.04 ^a^* **
Hb, g/dL	11.8	12.8	
median (IQR)	10.4- 13.1	11.5–14.0	** *0.03 ^b^* **
ANC, ×10^3^ cells/uL	3.1	3.6	
median (IQR)	1.3–6.9	2.3–4.6	0.57 ^b^
ALC, ×10^3^ cells/uL	1.4	1.6	
median (IQR)	0.9–3.5	1.0–2.4	0.92 ^b^
AMC, ×10^3^ cells/uL	0.3	0.5	
median (IQR)	0.07–0.5	0.3–0.6	** *0.02 ^b^* **
Platelets × 10^9^ count/uL	183	199	
median (IQR)	98–283	138–311	0.42 ^b^
C-RP, IU/L	4.5	1.3-	
median (IQR)	1.2–11.4	0.5–3.2	** *0.002 ^b^* **
LDH, IU/L	221	197	
median (IQR)	183–273	156–246	0.15 ^b^

^a^ based on Fisher-exact test, ^b^ based on Mann–Whitney test. Results are reported as median and interquartile range (IQR). Abbreviations: Hb, hemoglobin; ALC, absolute lymphocyte count; AMC, absolute monocyte count; ANC, absolute neutrophil count; uL, micro-liter; IU/L, international units per liter; LDH, lactate de-hydrogenase; C-RP, C-reactive protein; NA, not available.

**Table 4 cancers-14-01173-t004:** Results of regression analysis of clinical outcomes of COVID-19 in hematological patients following PSM.

Clinical and Laboratory Parameters	30-Day Mortality AOR (95% CI)	Hospitalization for Intensive Cure AOR (95% CI)	14-Day Viral Clearance AOR (95% CI)
Age >70 years	0.01 (−0.49–0.51)	−0.11 (−0.49–0.26)	0.1 (−0.26–0.48)
Sex: female	−0.01 (−0.44–0.42)	−0.16 (−0.49–0.17)	0.11 (0.21–0.43)
Not active and progressing disease	−0.64 (−1.09–0.2)	−0.12 (−0.44–0.20)	0.13 (0.19–0.45)
Charlson comorbidity index 0	−0.39 (−1.02–0.23)	−0.36 (−0.88–0.14)	0.38 (−0.12–0.89)
Lymphoid neoplasm	0.08 (−0.39–0.55)	0.037 (−0.30–0.38)	0.16 (−0.19–0.52)
Myeloid neoplasm	−0.18 (−0.7–0.34)	−0.11 (−0.49–0.26)	0.19 (−0.18–0.56)
Plasma cell neoplasm	0 (−0.42–0.28)	0 (−0.42–0.36)	0 (−0.42–0.34)
LDH > 243 IU/L	−0.0008 (−0.45–0.45)	−0.07 (−0.42–0.28)	0.32 (−0.04–0.67)
C-RP > 5 IU/L	−0.03 (−0.42–0.36)	−0.27 (0.57–0.03)	0.04 (−0.27–0.35)
AMC < 0.4 × 10^3^ cells/uL	0.68 (0.14–1.21)	0.17 (−0.14–0.49)	0 (−0.31–0.31)

Abbreviations: LDH, lactate de-hydrogenase; C-RP, C-reactive protein; AMC, absolute monocyte count, uL, microliter, IU/L, international units per liter.

**Table 5 cancers-14-01173-t005:** Univariate analysis of median overall survival based on main pre-infection clinical and laboratory parameters in unvaccinated hematological patients with COVID-19.

Clinical and Laboratory Parameters		N	mOS	95% CI	*p*-Value ^a^
Age, years	≤70	35	11.9	12.2–14.5	0.33
	>70	51	10.8	9.8–13.5	
Sex	Male	43	11.8	11.2–14.3	0.91
	Female	43	11.4	11.1–13.8	
Disease activity	Inactive	59	11.1	10.3–11.8	** *0.003* **
	Active or progressing	27	8.5	6.4–10.7	
Hematological malignancy	Myeloid	29	11.0	9.6–12.3	0.79
	Lymphoid	21	10.1	8.5–11.6	
	Plasma cell	36	10.4	9.2–11.7	
ANC, ×10^3^ cells/uL	≤4.0	60	9.9	8.8–11.0	0.06
	>4.0	26	11.6	10.8–11.3	
ALC, ×10^3^ cells/uL	≤1.0	40	6.7	5.8–7.6	0.27
	>1.0	46	10.8	9.9–11.8	
AMC, ×10^3^ cells/uL	≤0.4	43	9.6	8.3–11.0	** *0.03* **
	>0.4	43	11.1	10.3–12.0	
Pre-infection LDH	≤243 IU/L	63	10.7	9.8–11.6	0.31
	>243 IU/L	25	9.8	8.2–11.5	
Pre-infection C-RP	≤5 IU/L	62	10.7	9.8–11.6	0.31
	>5 IU/L	24	5.8	4.8–6.8	
Hospitalization for COVID-19	No	64	11.4	10.9–12.0	** *<0.0001* **
	Yes	22	7.6	5.5–9.8	
Viral clearance at 14 days	No	47	11.2	10.5–11.8	** *0.005* **
	Yes	39	8.4	6.2–10.6	
Seroconversion *	No	24	6.5	5.3–7.7	** *0.05* **
(* evaluated in N = 66)	Yes	42	12.0	12.0	

^a^ based on Kaplan–Meier plots, *p*-value was considered significant if <0.05 and indicated in bold italic. Abbreviations: ALC, absolute lymphocyte count; AMC, absolute monocyte count; ANC, absolute neutrophil count; uL, micro-liter; IU/L, international units per liter; LDH, lactate de-hydrogenase; C-RP, C-reactive protein. * is to highlight that only 66 patients were evaluated for seroconversion.

**Table 6 cancers-14-01173-t006:** Multivariable analysis of median overall survival based on main pre-infection clinical and laboratory parameters found significant in univariate analysis in hematological patients with COVID-19.

Covariate	HR (95% CI)	*p*-Value ^a^
Absolute monocyte count <400 cells/uL	8.66 (1.72–43.44)	** *0.009* **
Active and progressing disease	6.02 (1.62–22.44)	** *0.008* **
Hospitalization for COVID-19	7.14 (1.89–27.12)	** *0.004* **
Viral clearance at 14 days	4.26 (1.15–15.75)	** *0.03* **

^a^ hazard ratios and 95% confidence intervals were estimated with Cox regression analysis. Abbreviations: uL, microliter, HR, hazard ratio; CI, confidence interval.

## Data Availability

The original contributions presented in the study are included in the article. Further inquiries can be directed to the corresponding authors.

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
