# Peer review of "Reduced Absolute Count of Monocytes in Patients Carrying Hematological Neoplasms and SARS-CoV2 Infection"

_cancers, 2022, doi:10.3390/cancers14051173_

Round 1

Reviewer 1 Report

The article entitled "Absolute count of monocytes as predictor of reduced overall survival in patients affected by COVID-19 and hematological neoplasms"  presents the significance of simple routine parameters such as absolute count of monocytes as predictors of the overall survival in patients suffering from hematologic malignancies (three groups, with different disease each)  and COVID-19.

Although the number of the patients is low, an extensive statistical analysis has been done and revealed the significance of the above mentioned parameters.

Major comments:

1.The number of the patients included in the study is small, considering that the population is heterogeneous  with three different hematologic malignancies.

2.The aim of the study is not clearly defined.

3.The title does not seem to match with the conclusion. The title gives the impression that the AC of monocytes predicts the outcome of patients with COVID-19 and hematologic malignancies but the conclusion refers to the prognostic significance of the same parameters on the overall survival of the above patients, independently from the clinical course of COVID-19, which is rather confusing.

4.The prognostic significance of the monocytes is a statistical finding which needs further discussion.

5.Following the above conclusion the question is: what is the prognostic significance of the same parameters on the overall survival of patients with the same hematologic malignancies without SARS-CoV-2 infection?

Suggestions to the authors:

The authors should present more clearly their study aims and rephrase the title in accordance with the conclusions.

They should discuss possible explanations of the prognostic significance of the monocytes as well as any potential clinical intervention.

Finally  they should argue on the necessity to include or not in the study a control arm with non-COVID-19 hematologic patients.

Author Response

Dear Reviewer 1,

thanks for your suggestions. Please find in attachment a modified version of our manuscript.

The number of patients has been increased from 82 to 120, including 34 cases of COVID-19 in fully vaccinated patients.

We better clarified the aim of the study as follows: The primary endpoint was 30-day all-cause mortality (infection, progressive dis-ease, other) among fully vaccinated patients affected by hematologic malignancy test-ed positive to SARS-CoV-2 compared to the cohort of unvaccinated hematological pa-tients after PSM adjustment for baseline clinical variables. Secondary endpoints in-cluded rates of hospitalization in intensive care unit and viral clearance at 14 days, in fully vaccinated, compared with unvaccinated patients with hematological neoplasms after PSM adjustment for baseline clinical variables.

The title has been changes and the manuscript entirely re-written (highlighted in red in the enclosed version of the revised manuscript).

We added an important reference in the field (Maia, Haematologica 2020) which first disclosed, in a large series of COVID-19 patients including those with hematological neoplasms that compared to COVID-19 patients without hematological cancer, patients carrying he-matological neoplasms have decreased percentages of classical monocytes, immuno-regulatory natural killer cells, double-positive T cells, and B cells, that could compromise an initial response to the infection. Several studies, referred in the introduction, have disclosed that increased counts of monocytes (and their subclasses) are associated to clinical outcomes. Thus, we have discussed this point more in details, as suggested.

Based on observations reported above, and thanks to the introduction of a further control group (namely fully vaccinated patients who developed COVID-19), in which we confirmed the contribution of AMC<400 cells/uL in 30-day mortality adjusted odds ratio, we do not justify to modify the study design to include non-COVID-19 hematologic patients.

The manuscript (including introduction, results and discussion) has been entirely re-written and edited for proper English language, grammar, punctuation, spelling, and overall style.

Tables have been edited, and sentences to describe results have been re-written (highlighted in red in the enclosed version of the revised manuscript); we have included more explanations of the abbreviations below the tables.

In the future, reduced absolute counts of monocytes should be used as alert in any effective prevention strategy in contrasting severe/critical forms of COVID-19 in patients with hematological malignancies, also when the full vaccination cycle has been completed.

Reviewer 2 Report

In this manuscript, Authors analyzed absolute count of monocytes as predictor of reduced overall survival in patients affected by COVID-19 and hematological neoplasms. The manuscript is potentially interesting, but requires a major revision.

1. The manuscript should be edited for proper English language, grammar, punctuation, spelling, and overall style. For example:

- Summary: COVID-19 (COronaVIrus Disease-2019 – without capital “O” nad “I”) is a complex disease with variable clinical presentation and outcome and several (without bold) biomarkers have been evaluated that can allow us to predict an initial severe presentation or a critical evolution of the disease

- Introduction: COVID-19 (COronaVIrus Disease-2019) – the same as above

 - “In hematological patients, immunodeficiency is also caused by specific characteristics of the malignancies: among all hematological malignancies, patients who die because of infections, are more frequently affected by multiple myeloma, chronic lymphocytic leukemia and indolent non-Hodgkin lymphoma.”

This sentence is too long, I think it would be better to divide it into two sentences.

2. Results:

In my opinion, the results are presented not very clearly, in the text they are exactly the same as in the tables. Some sentences are imprecise and misleading.
For example:

“A total of 82 patients affected by hematological neoplasms and COVID-19 diagnosis were evaluated, including 29 (35%) with asymptomatic COVID-19 and 22 (27%) requiring hospitalization.”

“At time of follow-up, 64 patient (78%) were completely recovered, including 60 (73%) patients who obtained viral clearance, defined as two consecutive negative nasopharyngeal swabs, 2 (2%) had long term complications (disease progression and development of hemolytic anemia) and 15 (18 %) died.”

3. Tables:

There is no explanation of the abbreviations below the tables.

4. Conclusions:

How can these results be used in the future for this patient group? Any suggestions?

Author Response

Dear Reviewer 2,

thanks for your suggestions. Please find in attachment a modified version of our manuscript.

The manuscript (including introduction, results and discussion) has been entirely re-written and edited for proper English language, grammar, punctuation, spelling, and overall style.

Tables have been edited, and sentences to describe results have been re-written (highlighted in red in the enclosed version of the revised manuscript); we have included more explanations of the abbreviations below the tables.

In the future, reduced absolute counts of monocytes should be used as alert in any effective prevention strategy in contrasting severe/critical forms of COVID-19 in patients with hematological malignancies, also when the full vaccination cycle has been completed. 

Round 2

Reviewer 1 Report

The paper has been adequately revised and should be accepted for publication

Reviewer 2 Report

The modified version of this manuscript is clearer, and I accept it for publication.